# Sensing Method Using Dielectric Loss Factor to Evaluate Surface Conditions on Polluted Porcelain Insulator

**DOI:** 10.3390/s22239442

**Published:** 2022-12-02

**Authors:** Ľuboš Šárpataky, Bystrík Dolník, Ján Zbojovský, Uwe Schichler, Oliver Pischler, Bernhard Schober

**Affiliations:** 1Department of Electric Power Engineering, Faculty of Electrical Engineering and Informatics, Technical University of Košice, 04200 Košice, Slovakia; 2Institute of High Voltage Engineering and System Performance, Graz University of Technology, 8010 Graz, Austria

**Keywords:** dielectric loss factor, frequency spectrum, leakage current, pollution, porcelain insulator

## Abstract

Insulator diagnostics is still a topical issue. No one has yet found out how to accurately determine the condition of all insulators and decide when replacement or maintenance is required. Insulators are one of the main components of a transmission and distribution system and must withstand high voltages in all weather conditions. Moisture and dirt are the main factors influencing the insulating properties of insulators. This article deals with the effect of pollution on a porcelain insulator. An Omicron MI 600 measuring system monitors the changes in the dielectric loss factor and leakage current in a wide frequency range (10 Hz to 1 kHz) to evaluate the contamination level. We applied three high voltage levels (5 kV, 7.5 kV, and 10 kV) to the porcelain insulator to monitor changes in the mentioned quantities with various frequencies. The measurement results confirmed the usability of the dielectric loss factor and leakage current for the diagnosis of insulator pollution. The dielectric loss factor showed more promising results than the leakage current.

## 1. Introduction

Insulators belong to fundamental insulation components in transmission and distribution lines, and they must meet specific requirements. Their basic parameters are dielectric strength, mechanical strength, and surface conductivity. The respective values must be maintained throughout the insulator’s lifetime. For commissioning purposes, manufacturers test insulators to decrease the failure rate. They are monitored during their operation using various diagnostic methods to achieve the highest possible reliability [1,2,3].

Because companies manufacture insulators mainly for outdoor use, weather conditions and pollution influence their dielectric strength and mechanical strength. In different areas, various environmental effects influence insulators, reducing their dielectric strength and thus endangering the reliability of the electricity supply. Various insulator types were developed and tailored for specific environments to prevent blackouts. However, it is not always possible to eliminate all threats in a given environment. Therefore, new diagnostic methods that can determine the critical condition of an insulator surface before it fails are still being researched and developed. Thanks to research and development, maintenance can be better planned, and thus the reliability is increased [2,4].

The unfavorable effects of electrical aging strongly depend on the nature and type of electrical insulation system. Electrode size, insulation thickness, the insulator’s surface condition, the maximum intensity and frequency of applied voltage, and the failure rate affect the magnitude of electrical stress. Another factor affecting insulator degradation is pollution. Increasing humidity can multiply the effect of pollution. If the insulator is extremely dirty, the operating voltage can cause problems with the external insulation due to partial discharges or overheating. Erosion can occur on the insulator surface under partial discharge activity, leading to material inhomogeneities on the surface. On the eroded surface, a crack subsequently develops. Pollution can quicken the erosion process [5,6,7,8].

The factors that affect the dielectric strength of the insulator the most are pollution and humidity. The combination of those two factors can significantly reduce the dielectric strength, cause an increase in active losses, and reduce the lifetime of insulators. The leakage current increases with increasing surface contamination. Especially, leakage current is higher on a wet, contaminated layer than on a dry insulator, which over a long time can cause the insulator to deteriorate and eventually fail, leading to a flashover or breakdown [2,4,9].

The study and diagnostics of insulators are still very current topics. Researchers develop new techniques and ways to determine the condition of an insulator as accurately as possible. Except for the visual control and measurement of the electric field, the fundamental diagnostic methods include measuring the leakage current. The leakage current can determine the influence of humidity and pollution. This quantity can be used to diagnose all used materials, such as porcelain, glass, or polymers, and indicate how the condition of the surface changes. Many researchers have used leakage currents to analyze more types of pollution. Some researchers compare various pollutants or use the same pollution type with different amounts of active substances. Others use the salt fog method, included in the standard IEC-61109:2008 [10]. This method uses a chamber to ensure the uniformity of the contamination on the entire surface and a clean comparison between different levels of the amount of salt [11,12,13,14,15]. Another method used is pre-contamination with a set solution of salt, water, and kaolin. The method of pre-contamination helps to investigate the dry pollution layer and hydrophobicity. By gradually increasing the humidity, it is possible to track how the leakage current increases and identify the critical humidity level, which is the point from which it increases drastically [16,17,18,19]. Research is also devoted to other types of pollution and different additives that can be used instead of salt and kaolin [20,21,22,23].

One of the leakage current measurement methods is performing harmonic analysis. Harmonics are contortions of the sinusoidal electrical waveform that can be evaluated by total harmonic distortion (THD) or by the individual components. THD is defined as the ratio of the sum of the powers of all harmonic components to the power of the fundamental frequency. Research has shown that humidity and pollution affect the harmonic components. Humidity mainly affects the first (fundamental) harmonic component, and pollution affects the third, fifth, and seventh harmonic components. When the degree of pollution increases, the ratio of the third, fifth, and seventh harmonic components changes [24,25,26,27,28,29,30].

The usage of numerical methods has increased. Researchers use them to simulate various faults or the condition of an insulator’s surface. The BEM (boundary element method) and FDM (finite difference method) are occasionally employed in addition to the well-known FEM (finite element method). These numerical methods are used individually or in combination with experimental measurements, where the measured results are subsequently compared [31,32,33,34,35].

Electrical power engineering utilizes the dielectric loss factor for transformers, cables, winding, and capacitor diagnostics. Spectroscopy is used on transformers to analyze the degradation of the transformer’s insulation system. Changes in the dielectric loss factor in different parts of the frequency range indicate insulation aging [36,37]. Conductivity and power cable polarization loss values can be obtained by using low-frequency measurements of the dielectric loss factor [37,38]. Some researchers apply dielectric loss factor measurement to study the aging of insulators. Results show that aging correlates with dielectric loss factor values. Research has pointed out that insulation degradation can be measured using the dielectric loss factor [39,40,41,42].

Various studies examine an insulator’s surface. In assessing insulator surface contamination, the leakage current has demonstrated positive results. The development of other diagnostic methods is still in progress, and many seem to be applicable for determining the contamination level of the insulator’s surface.

We compare the dielectric loss factor as a contamination level indicator with the leakage current in this research. According to the literature review, few researchers have used the dielectric loss factor as a diagnostic quantity for insulators. We applied three high voltage levels to the insulator to measure the leakage current and dielectric loss factor. Measurements have been performed in a wide frequency range to research how frequency influences the dielectric loss factor values for different pollution levels. The measurements were carried out at various frequencies with high voltages from 5 kV to 10 kV. Two solutions for contamination of the insulator’s surface were created by mixing kaolin, salt, and water. We measure the leakage current simultaneously with the dielectric loss factor to compare and assess the measurement correctness. The dielectric loss factor shows satisfying results as a quantity useable for pollution level classification. Moreover, the leakage current seems less sensitive to moisture and contamination changes.

## 2. Materials and Methods

Research and measurements took place in a high-voltage laboratory at the Graz University of Technology. Measurements were performed on a new U 70 BL cap-and-pin porcelain insulator. The insulator is suitable for transmission lines as a part of an insulator string, and its surface is glazed. Figure 1 depicts the porcelain insulator standardized according to the IEC 60305 standard [43]. The diameter (D) is 255 mm, and the spacing (H) is 146 mm. The U 70 BL porcelain insulator guarantees a lightning impulse withstand voltage of 100 kV in dry conditions, a power frequency withstand voltage of 70 kV in dry conditions, and a power frequency withstand voltage of 40 kV in wet conditions.

The ambient temperature ranged from 22 °C to 25 °C during the measurements. These temperature changes cannot significantly affect the measurement results.

Firstly, we prepared solutions for the contamination of insulators. We mixed two different levels of pollution solutions (L1 and L2) using tap water, kaolin, and salt. The amount of kaolin was the same for both pollution levels, in which we added 40 g of kaolin for each liter of water. Varying the amount of salt changes the conductivity of the solution. The individual pollution levels, with the amount of salt added per liter of distilled water, and the corresponding conductivity values of the mixed solutions, according to IEC/TS 60815-1:2008, are in Table 1 [44].

The first measured case was the clean insulator. After the clean insulator measurements, we soaked the insulator in tap water and repeated the measurements. The soaking process took about a minute. The air between the insulator’s ribs at the bottom of the insulator’s skirt (petticoats) was removed by appropriate rotation and tilting in the water to ensure the soaking process of the whole insulator’s surface. Other performed measurements used the polluted insulator with dry and wet pollution layers (10 min after immersion of the insulator in an artificial pollution solution). The drying process of the pollution layer took 24 h after immersion in the solution.

Figure 2 shows a detailed description of the measurement procedure. Table 2 explains the abbreviations used in the flowchart in Figure 2.

We measured mentioned quantities simultaneously. We connected the high voltage to the insulator’s pin and grounded the cap. Figure 3 shows the measurement setup.

An FG250D function waveform generator (G) and Proline 3000 amplifier (►) were connected to the insulator’s pin through a transformer. We used 5 kV, 7.5 kV, and 10 kV voltage levels, and a wide range of frequencies of the testing voltage were applied to the insulator. The measurement frequency range started at 10 Hz, and the frequency was increased stepwise to 1000 Hz. Firstly, we used a 10 Hz step until we reached a 100 Hz level. After we reached 100 Hz, we changed the measurement step to 50 Hz. A personal computer (PC) connected to the measuring devices via a USB port controlled the measurement and recorded the dielectric loss factor and leakage current every 0.9 s. The measured quantities were recorded on a storage medium using a PC. OMICRON software for the MI 600 instrument calculated the average leakage current and dielectric loss factor from 100 measured values, controlled the measurement, and stored the data. The reference capacitor manually set in the OMICRON software was connected to the MI 600 reference sensor. The reference capacitor had a capacity of 37.86 pF. We connected the porcelain insulator to the MI 600 test object sensor and sensors by fiber-optical cables to control station MCU 502. The measurement diagram of the leakage current and dielectric loss factor is shown in Figure 4.

The dielectric loss factor is a tangent of the dielectric loss angle (*δ*) between two vertical components, namely, the capacity current *I*_C_ and the resistance current *I_R_*.
(1)$$\tan\delta =\frac{I_{R}}{I_{C}}=\frac{\frac{U}{R}}{\omega CU}=\frac{1}{\omega CR}$$

The dielectric loss factor can be calculated by Formula (1). Our measurement system directly evaluated the value of the dielectric loss factor. The Omicron MI 600 estimates the dielectric loss factor based on the phase shift between the measured current and the applied voltage on the sample. It compares the measured sample with the reference sample.

## 3. Results

Test measurements were carried out following immersion in the solution to determine the stabilization of the measured quantities on the polluted layer. Immediately after soaking in the solution, partial discharges occurred on the surface of the insulator, which affected the measured quantities. Therefore, we measured quantities on a wet layer 10 minutes after soaking. Figure 5 shows the dependence of quantities on time at a voltage of 5 kV for both the first and second levels of pollution. We chose three different measuring frequencies. The dielectric loss factor gradually increases at various frequency levels. The highest values of the dielectric loss factor, when comparing measurements between frequencies, were measured at a frequency of 10 Hz. As the frequency increases, the dielectric loss factor decreases. Comparing the dielectric loss factor at different pollution levels, the shapes of the curves are similar. Measurements at the second pollution level result in higher dielectric loss factor values than those at the first. By comparing the measurement results at the 1st, 10th, and 15th minutes, we found that the dielectric loss factor decrease between the 10th and 15th minutes was 5.8% on average.

The leakage current had a gradual development when changing the frequency. The leakage current reached the highest values at the frequency of 250 Hz. It stabilized similarly to the dielectric loss factor. When comparing the leakage current, the decrease between the 10th and 15th minutes was 11.5%. Figure 5 depicts the measured data.

We investigated the frequency dependencies for a clean insulator (L0) and two levels of artificial pollution (L1, L2) by measuring the dielectric loss factor and the leakage current. The used frequency range was from 10 Hz to 1 kHz. Research also deals with the influence of the voltage level (5 kV, 7.5 kV, 10 kV) and whether the insulator surface is dry or wet. We used the Matlab software to plot the dielectric loss factor and leakage current dependencies.

### 3.1. Dielectric Loss Factor Measurement Result

Firstly, we compared the measurements at different voltages. Figure 6 shows the dependences of the dielectric loss factors on the frequency at different voltage levels. To make the results more intelligible, we divided the measurements on a clean insulator into two graphs.

By comparison of measured values, the change in the dielectric loss factor, depending on the voltage level, can be evaluated. Table 3 shows data representing the percentage difference between measurements at various voltages. The table also points out how the voltage affects the dielectric loss factor depending on the insulator’s surface (clean–polluted, dry–wet). Figure 6 shows that the change in voltage level does not significantly influence the dielectric loss factor. The measurement results on a clean insulator showed that the dielectric loss factor differs only minimally with voltage change, and the curves are almost identical in shape. In Figure 6a, the dielectric loss factor measured at a 10 kV voltage level is slightly higher those at voltage levels of 5 kV and 7.5 kV. The average dielectric loss factor difference between voltage levels in percentage was 0.26–2.3%. The maximal difference between single values was 4.1% at a frequency of 10 Hz, comparing 7.5 kV and 10 kV voltage levels. Figure 6b represents measurements on the wet, clean insulator. The average difference is higher than for the dry insulator and varies from 1.52–2.1%. At the frequency of 40 Hz, the highest single value difference reaches 6.7% at applied voltages of 7.5 kV and 10 kV.

Measurements of the insulator contaminated by the first level of artificial pollution (L1) show a difference between measurements at the dry and wet pollution layers. Measurement on the insulator with a dry pollution layer shows almost identical results for all voltage levels. In Figure 6c, the measurements on the dry, polluted insulator show an average difference between dielectric loss factors at various voltages of 1.05–2.22%. The maximal difference between single values was 3.94% at a frequency of 40 Hz between 5 kV and 10 kV voltage levels. The difference between dielectric loss factors at applied voltage levels increases when measuring on the wet, polluted insulator. From the frequency of 10 Hz to the frequency of 300 Hz, the dielectric loss factor increases with rising voltage. From 350 Hz, measured values for voltage levels of 7.5 kV and 10 kV were almost identical again. The curves are not smooth, as for the dry pollution layer, but the trends of the curves are similar. For the measurement of the wet, polluted insulator at the first pollution level, the average difference between dielectric loss factors in percentage is from 9.76% to 26.8%. The maximal difference between single values reaches 37.38%. It occurs at a frequency of 800 Hz between 5 kV and 10 kV.

Measurement of the insulator contaminated by the second level of artificial pollution (L2) shows the same trend of the dielectric loss factor comparing dry and wet pollution layers as for the first pollution level (L1). The insulator contaminated by the dry pollution layer performed almost identically for all voltage levels. Figure 6d depicts the measurement of the dry, polluted insulator. The average dielectric loss factor difference at all applied voltage levels was 0.93–1.31%. The maximal difference was 3.58% at a frequency of 30 Hz for 7.5 kV and 10 kV voltage levels. There is an increase in the dielectric loss factor comparing the first and second pollution layers when the layer is wetted. The average difference varies from 8.83% to 20.26%. At a frequency of 450 Hz between 5 kV and 10 kV, voltage levels show a maximal difference between single values of 31.92%. All average dielectric loss factor differences are displayed in Table 3. This table compares how the voltage level affects the dielectric loss factor.

Secondly, we compare measurements at different pollution levels and the clean insulator. Figure 6 shows the dependences of the dielectric loss factors on the frequency at different pollution levels and the same voltage level. 

To evaluate dielectric loss factor differences between pollution levels, we used relative values to the maximal value, measured at a frequency of 10 Hz for the second pollution level. The highest dielectric loss factor value occurred at the same frequency and the same contamination for all applied voltages.

It can be seen from the individual graphs in Figure 7 that the dielectric loss factor has a decreasing trend at lower frequencies. There is a minimal difference between measurements on a clean insulator, and the influence of soaking in clean water is negligible. On the other hand, when measuring on a contaminated insulator, the difference between the measurement on a dry layer of pollution and a wet layer of pollution is considerable. Even though the measurement took place 10 min after soaking in the solution and applying voltage, the difference between the dry and wet layer measurements was significant, between 33.8–54.2%. The higher the voltage, the higher the difference between the dry and wet conditions. If we compare the measurements with a dry layer of pollution, the most significant difference occurs at low frequencies. The courses of the curves are parallel, resulting in a stepwise increase in the dielectric loss factor. This applies to the entire frequency spectrum. The average difference between individual dry pollutions was in the range of 23.7–25.1%. By comparison of measurements on the wet pollution layer, the curves are parallel, and the dielectric loss factor differences are significant until the frequency reaches 200–300 Hz. At higher frequencies, the shapes of the curves zigzag, and the difference between them changes; sometimes, they are almost identical. From the frequency of 400 Hz, the dielectric loss factor on the wet layer of pollution starts to increase (a little earlier at the second level of pollution), which could be caused by polarization effects. They manifest themselves in these conditions in the given frequency range. The average difference, appearing mainly at low frequencies, is 21.3–14.5%. All average differences in the dielectric loss factors are shown in Table 4.

### 3.2. Leakage Current Measurement Result

The leakage current was measured simultaneously with the measurement of the dielectric loss factor. The process of calculations and assessments was the same as for the dielectric loss factor.

When measuring on a clean insulator, it is clear that as the voltage increases, the leakage current also increases linearly with the voltage (on a logarithmic scale). Soaking the insulator in water only minimally affected the magnitude of the leakage current. The curves are parallel, and the increase in leakage current is identical for both dry and wet measurements.

Measurement on the first level of pollution in Figure 8c shows that the leakage current increase is gradual with the voltage increase. The change occurs when measuring with a wet layer of pollution. From the frequency of 500 to 800 Hz, when the leakage current increases rapidly, the difference between the measurement on the dry layer and the wet layer is apparent.

When measuring the second pollution level at the voltage of 7.5 kV, the leakage current increases rapidly from 600 Hz. For voltage levels of 5 kV and 10 kV, a rapid increase starts at a frequency of 900 Hz. These changes and differences in the shapes of the curves could happen due to the ever-changing wet pollution layer and polarization effects. The average differences in leakage current between various applied voltage levels are shown in Table 5. The table shows a stable increase in leakage current with increasing voltage for dry and wet conditions with pollution.

When comparing the results in Figure 9, the differences between the individual measurements are not as significant as they were with the dielectric loss factor. The measured leakage current on wet polluted layers for both pollution levels grows faster, especially at higher frequencies. The most significant difference occurred when measuring at a voltage level of 7.5 kV. The curves for a wet measurement start to increase sharply from the frequency of 600 Hz. A lower difference occurs when applying voltage levels of 5 kV and 10 kV. However, the leakage current is higher than other curves considering frequencies of 900 Hz, 950 Hz, and 1000 Hz. Leakage current curves at these frequencies on wet layers of pollution are separated. The difference between wet pollution layers is minimal. The detailed view in the graphs points out that the individual measurements are graduated, but the difference between them is low. By comparison of average differences between the first and second degree of pollution, the dry layer differs from 1.33 to 2.07%. On the wet layer, the difference is from 1.88 to 2.97%. If we compare measurements on dry and wet layers, the differences are higher (5.7–9.9%) compared with different pollution level measurements. All details on such differences are shown in Table 6.

## 4. Discussion

The dielectric loss factor measurement shows low sensitivity to voltage changes. Despite a significant change in voltage, the change in the dielectric loss factor was minimal. The average differences in the dielectric loss factor were calculated to compare measurements at various voltage levels (Table 3). Only results on the wet pollution layer show an increase in the dielectric loss factor depending on the voltage level. On the dry pollution layer, changes in the dielectric loss factor were minimal, so voltage changes cannot affect the results. This statement is confirmed in [40], where the effect of voltage changes was also minimal. Comparing the dielectric loss factor at the same applied voltage showed that the dielectric loss factor is significantly dependent on the contamination of the insulator. The dielectric loss factor also depends on the wetting of the contaminated layer. The effect of soaking a clean insulator in tap water was noticeable, but the differences between measurements on contaminated insulators were considerably higher. In particular, the measurements on the dry, polluted layer for both pollution levels were simultaneous and comparable at any frequency. The highest differences in dielectric loss factor are found at lower frequencies. If we compare the measurements on wet, polluted insulators, the curves coincided at a frequency of around 100–200 Hz. The dielectric loss factor is used mainly for diagnostics of aging processes. Aging causes the deterioration of insulation, and our research corresponds with previous studies. In addition to aging, pollution worsens the insulation surface conditions, and the increase in the dielectric loss factor in our research is similar to values found in other studies [40,41,42]. Furthermore, there is an increase in the dielectric loss factor, which could be caused by the changing humidity and by polarization effects. The dielectric loss factor was higher for the entire frequency spectrum for the second pollution level compared to the lower pollution level.

An evaluation of the leakage current, measured simultaneously with the dielectric loss factor, was made for comparison. Many researchers frequently use the leakage current to point out the increase when an insulator’s surface deteriorates. Our measurements also confirmed that the leakage current depends on humidity and pollution. All new and older research refers to changes in leakage current with insulator surface deterioration. Contamination increases the leakage current, as well as the aging process. We confirm this fact in our experimental study [12,13,14,15,16,17,18,19,20]. However, the changes in the leakage current, comparing results at different pollution levels, are not so significant. The impact of the wetting process on the polluted layer was higher than the influence of the degree of pollution. The leakage current was much more sensitive to the voltage level than the dielectric loss factor. The change in the value of the leakage current is most pronounced at high frequencies.

## 5. Conclusions

Experimental measurements of the dielectric loss factor and leakage current in laboratory conditions were performed to investigate the possibility of using the dielectric loss factor to determine the contamination level of a porcelain insulator. Two artificial pollution levels as solutions were prepared and applied to the insulator. The study deals with the dielectric loss factor and the leakage current measurements in the frequency range from 10 Hz to 1 kHz. The application of three different high voltage levels compares the impact of voltage on measurement results. Electric power engineers use the dielectric loss factor mainly to examine the degradation and aging of insulations of cables, transformers, and generators. This article shows two new aspects of dielectric loss factor use:
1.Usage for diagnostics of insulators. Our measurements at high voltage confirm the usability of the dielectric loss factor for diagnostics of insulators.2.The dielectric loss factor can be used not only for aging process evaluation but has promising results to indicate contamination of insulators. Our measurements in a frequency spectrum widen diagnostic possibilities at various frequency levels. In some applications, it can help to avoid interference with the industrial frequency, which can reduce the accuracy of measurements.

The measurement results showed:
The dielectric loss factor is more sensitive to changes in the conditions of the insulator surface than the leakage current. When looking at the level of pollution, it showed more significant differences in both dry and wet conditions.Measuring the dielectric loss factor in the frequency spectrum or as a single frequency value is a suitable method for diagnosing contamination of the insulator surface.In dry conditions, the dielectric loss factor was more stable (with a lower standard deviation), with clearly identifiable changes. The advantage is that dry conditions are more suitable for diagnostics using the dielectric loss factor than a wet environment. In wet conditions, there are frequent changes in the parameters of the polluting layer, which can affect the measured quantities. On the contrary, the results of the leakage current measurements showed that the differences were less pronounced in dry conditions than in wet conditions for all pollution levels.

The next step of our experiments on polluted insulators will be performing measurements using different measuring systems. We will use various devices and different voltage levels to measure the dielectric loss factor in the frequency spectrum. We will apply an artificial aging process to the insulator to confirm the applicability of the dielectric loss factor as a pollution diagnostics tool for polluted and aged insulators. 

## Figures and Tables

**Figure 1 sensors-22-09442-f001:**
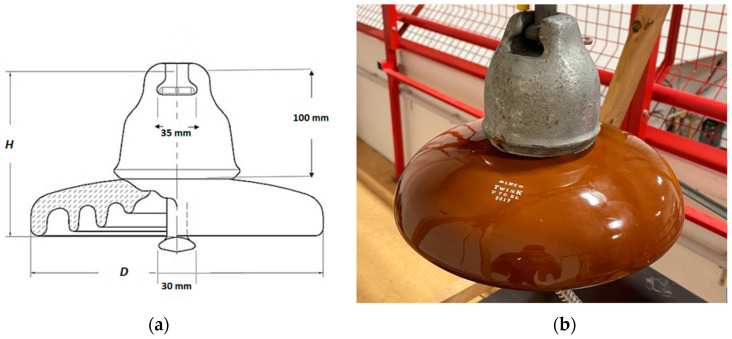
U 70 BL porcelain insulator: (**a**) Parameters; (**b**) Used insulator.

**Figure 2 sensors-22-09442-f002:**
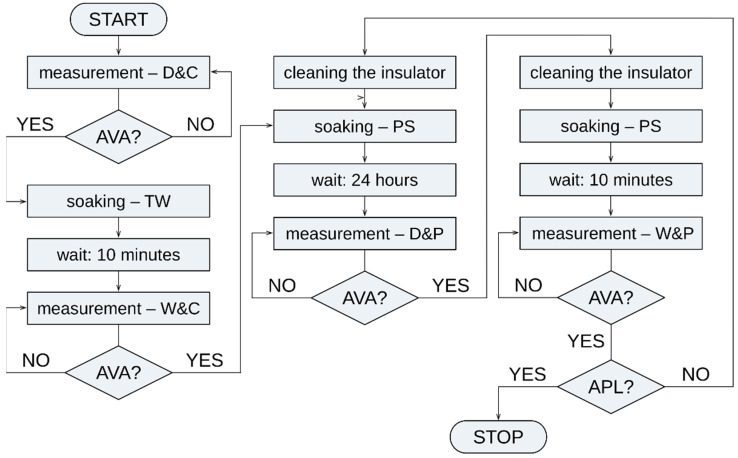
Flowchart of the measuring procedure.

**Figure 3 sensors-22-09442-f003:**
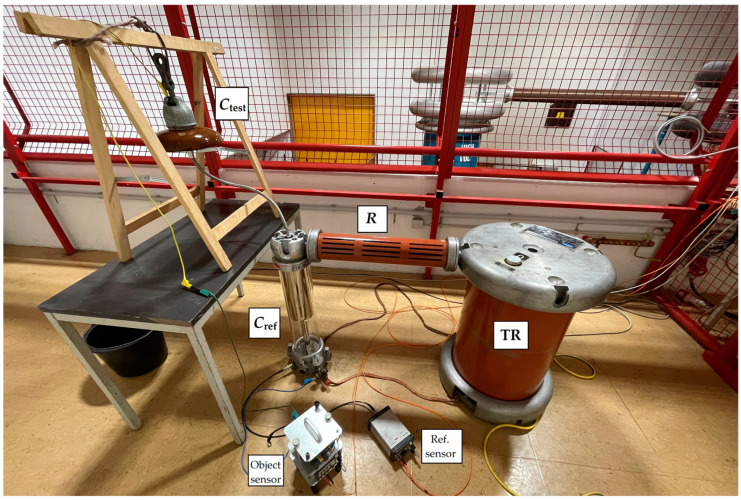
Measurement setup for measurement with OMICRON MI 600.

**Figure 4 sensors-22-09442-f004:**
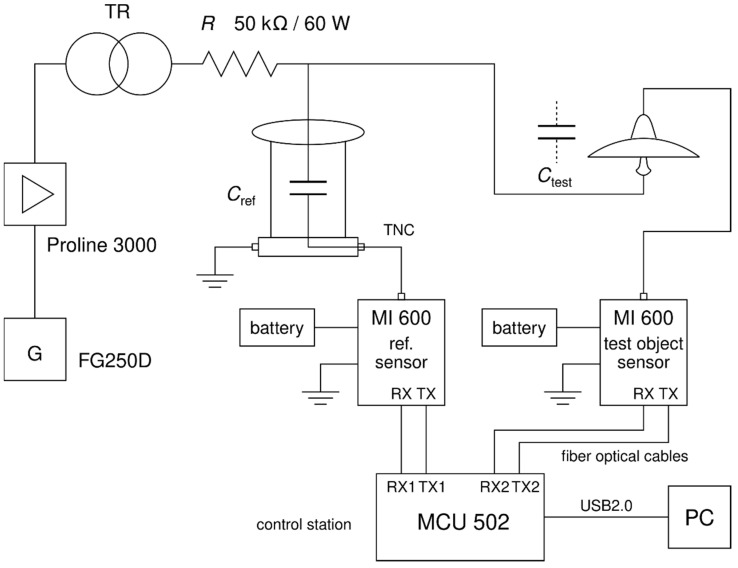
Measurement diagram for measurement with OMICRON MI 600.

**Figure 5 sensors-22-09442-f005:**
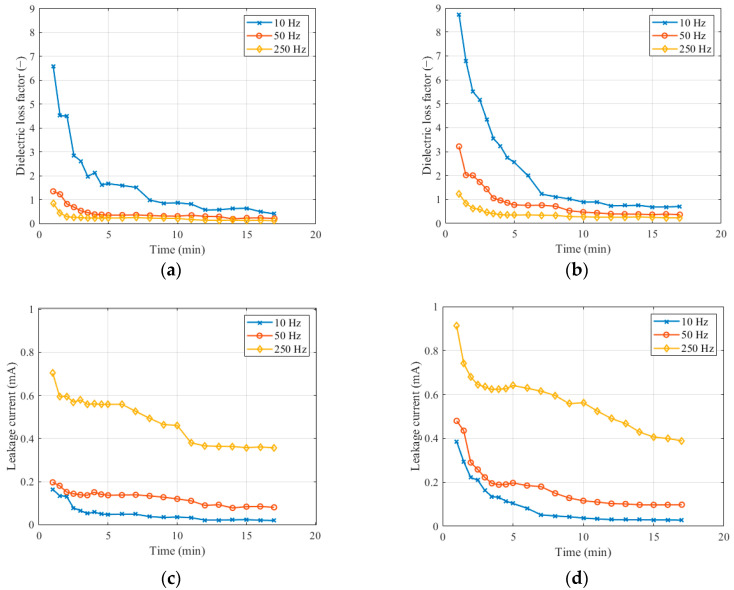
Change in quantities: (**a**) dielectric loss factor (L1); (**b**) dielectric loss factor (L2); (**c**) leakage current (L1); (**d**) leakage current (L2).

**Figure 6 sensors-22-09442-f006:**
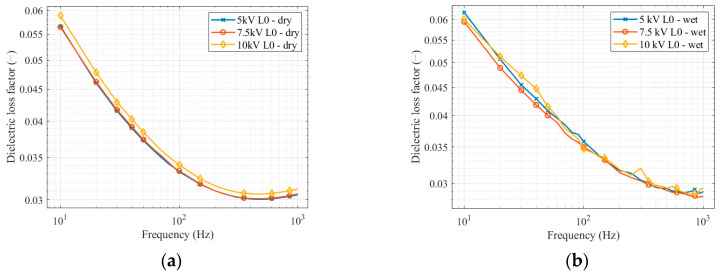
Dielectric loss factor frequency spectrum for different voltage levels: (**a**) clean insulator (dry); (**b**) clean insulator (wet); (**c**) first pollution level; (**d**) second pollution level.

**Figure 7 sensors-22-09442-f007:**
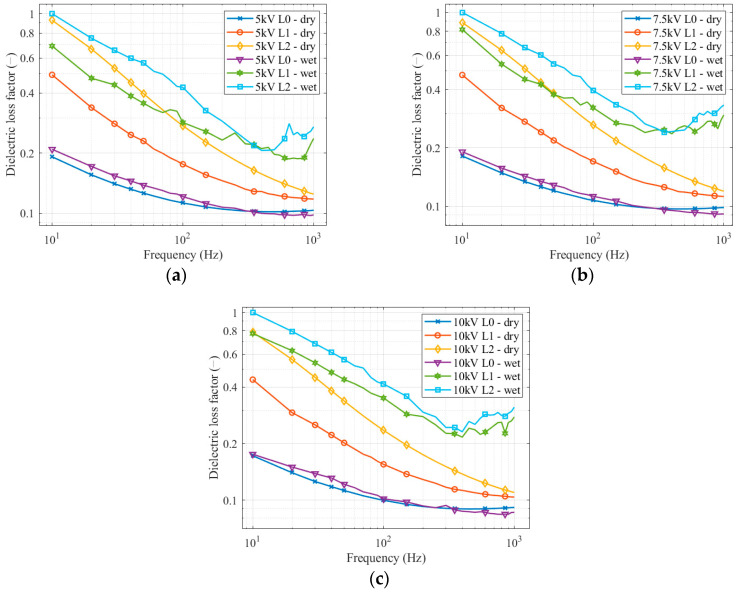
Dielectric loss factor frequency spectrum for a clean insulator and different pollution levels: (**a**) 5 kV; (**b**) 7.5 kV; (**c**) 10 kV.

**Figure 8 sensors-22-09442-f008:**
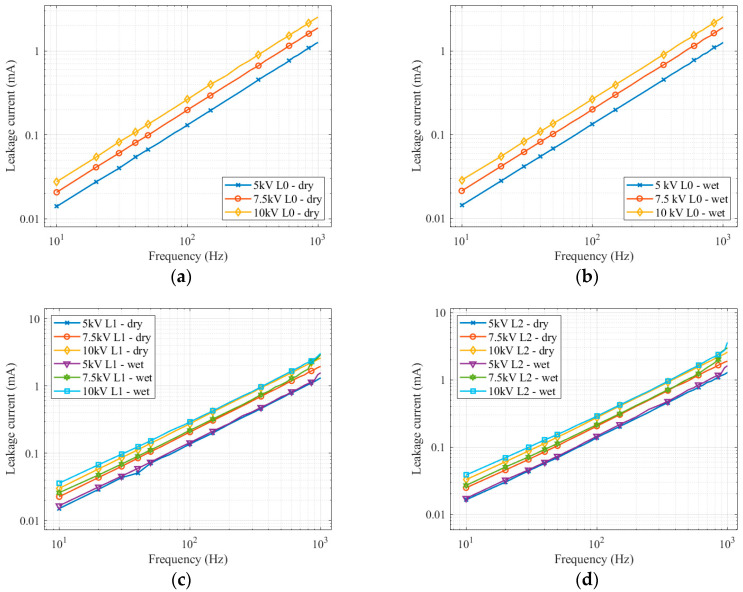
Leakage current frequency spectrum for different voltage levels: (**a**) clean insulator (dry); (**b**) clean insulator (wet); (**c**) first pollution level; (**d**) second pollution level.

**Figure 9 sensors-22-09442-f009:**
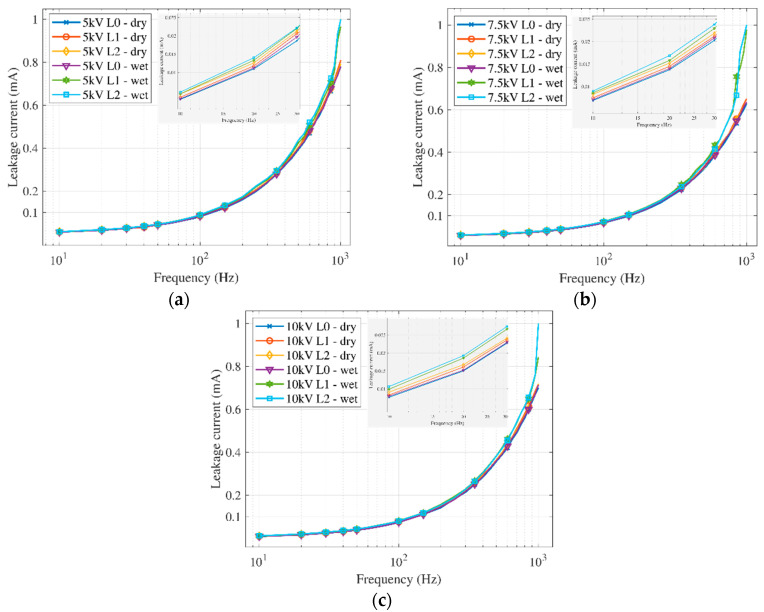
Leakage current frequency spectrum for the clean insulator and different pollution levels: (**a**) 5 kV; (**b**) 7.5 kV; (**c**) 10 kV.

**Table 1 sensors-22-09442-t001:** Information about salt amount and conductivity of artificial pollution levels L1 and L2.

Pollution Level	NaCl Amount (g/L)	Solution Conductivity (µS/cm)
L0	0	50
L1	6	129
L2	16	286

**Table 2 sensors-22-09442-t002:** List of abbreviations in the flowchart in Figure 2.

Acronym	Acronym Meaning
AVA	All Voltages Applied
APL	All Pollution Levels
D&C	Dry and Clean
D&P	Dry and Polluted
W&C	Wet and Clean
W&P	Wet and Polluted
TW	Tap Water
PS	Polluting Solution

**Table 3 sensors-22-09442-t003:** Comparison of differences between voltage levels for dielectric loss factor.

Pollution Level	Average Difference (%)		
	5–7.5 kV	5–10 kV	7.5–10 kV
L0 (dry)	0.26	2.03	2.08
L1 (dry)	1.05	2.22	1.18
L2 (dry)	0.93	1.29	1.31
L0 (wet)	1.69	1.52	2.1
L1 (wet)	18.9	26.8	9.76
L2 (wet)	12.96	20.26	8.83

**Table 4 sensors-22-09442-t004:** The average difference between dielectric loss factors of different pollution levels and different conditions.

Comparison	Average Difference (%)		
	Voltage 5 kV	Voltage 7.5 kV	Voltage 10 kV
L1 (dry)—L1 (wet)	38.7	49.8	54.2
L2 (dry)—L2 (wet)	33.8	41.1	47.0
L1 (dry)—L2 (dry)	25.1	25.0	23.7
L1 (wet)—L2 (wet)	21.3	15.8	14.5

**Table 5 sensors-22-09442-t005:** Comparison of differences in leakage current for various voltage levels.

Pollution Level	Average Difference (%)		
	5–7.5 kV	5–10 kV	7.5–10 kV
L0 (dry)	33.1	50.0	25.3
L1 (dry)	33.6	50.3	25.1
L2 (dry)	33.4	50.1	25.0
L0 (wet)	33.4	49.9	24.8
L1 (wet)	36.7	51.0	22.1
L2 (wet)	35.5	50.4	23.0

**Table 6 sensors-22-09442-t006:** The average difference between leakage current of both pollution levels and different conditions.

Comparison	Average Difference (%)		
	Voltage 5 kV	Voltage 7.5 kV	Voltage 10 kV
L1 (dry)—L1 (wet)	5.7	9.9	7.1
L2 (dry)—L2 (wet)	7.08	9.8	7.8
L1 (dry)—L2 (dry)	2.07	1.48	1.33
L1 (wet)—L2 (wet)	2.15	2.97	1.88

## Data Availability

The data supporting the reported results were recorded by the authors and can be obtained from the corresponding author.

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
