# Peer review of "Sensing Method Using Dielectric Loss Factor to Evaluate Surface Conditions on Polluted Porcelain Insulator"

_sensors, 2022, doi:10.3390/s22239442_

Round 1

Reviewer 1 Report

This manuscript focuses on the sensing method based on the dielectric loss factor to evaluate surface 2 conditions on polluted porcelain insulator. The contents are interesting, but some improvements should be considered as below:

1. The statement for Figure 4 in line 167 is not for Figure 4. And the comment in Figure 5 is too short. Discussion of results in Figure 5 should be discussed in detail..

2. The table 3, line 221. There are 2 columns 5kV – 7.5kV, I don't understand why there are 2 columns that are the same. I think this table needs more explanation.

3. In these 2 paragraphs (line 195 and 209), there are quite a lot of sentences and repetitive structures “Measurements of the insulator contaminated by the ...”. The other passages in the article also have the same structure and sentences, reused many times

4. Section 4 (Discussion and conclusions): There is no discussion of your results in comparison to others. I believe the authors should continue their discussion.

5. In the conclusions, the main results and the innovation should be highlighted in detail.

Reviewer 2 Report

This paper investigated polluted porcelain insulator diagnostics using the dielectric loss factor. The author gives a detailed introduction and description of the experiments. There are some suggestions for the author:

1.       It is better to illustrate the degradation mechanism of the porcelain insulator briefly;

2.       Provide clearer pictures in Figure 2 and Figure 4;

3.       Why was the NaCl solution can be equal to the environmental pollution for the porcelain insulator?

Please put the formula that the dielectric loss factor is calculated from the measurement results.

Round 2

Reviewer 1 Report

The revised manuscript is significantly improved and addresses all of the reviewers' comments.